# The Effect of Parasitization by Trichodinid Ciliates on the Mortality of Cultured Pacific Fat Sleeper larvae (*Dormitator latifrons*)

**DOI:** 10.3390/ani14203037

**Published:** 2024-10-20

**Authors:** Byron Manuel Reyes-Mero, Yanis Cruz-Quintana, Rossanna Rodríguez-Canul, Enric Gisbert, Ana María Santana-Piñeros

**Affiliations:** 1Grupo de Investigación en Sanidad Acuícola, Inocuidad y Salud Ambiental (SAISA), Departamento de Acuicultura, Pesca y Recursos Naturales Renovables, Facultad de Acuicultura y Ciencias del Mar, Universidad Técnica de Manabí, Bahía de Caráquez 130104, Ecuador; byron.reyes@utm.edu.ec (B.M.R.-M.); yanis.cruz@utm.edu.ec (Y.C.-Q.); 2Unidad Merida, Departamento de Recursos del Mar, Centro de Investigación y de Estudios Avanzados del IPN, Km 6 Carretera Antigua a Progreso, Mérida 97310, Yucatán, Mexico; rossana.rodriguez@cinvestav.mx; 3Aquaculture Program, Institute of Agrifood Research and Technology, IRTA, Crta. Poble Nou Km 5.5, 43540 La Ràpita, Spain; enric.gisbert@irta.cat

**Keywords:** protozoan, parasite, pacific fat sleeper, larvae, air-breathing fish, infestation rate

## Abstract

**Simple Summary:**

The cultivation of the Pacific fat sleeper (*Dormitator latifrons*) is of commercial interest in the Eastern Central Pacific. Its domestication potential is affected by the presence of pathogens. This study aimed to identify two trichodinid ectoparasites that are associated with *D. latifrons* larval mortality. We evaluated a total of 4320 fish larvae, 1080 fish larvae per treatment, revealing the presence of *T. acuta* and *T. compacta* in one treatment. Larval mortality in the infected treatment reached up to 58% by the fourth day post-hatching, at 28 °C, due to *Trichodina* infestation, resulting in significantly lower survival in the infected tanks compared to the uninfected tanks. This is the first geographical record of these protozoan parasites in a species that is native to the East Central Pacific Basin and intended for culture.

**Abstract:**

Trichodinidae, a prevalent group of protozoan ectoparasites in aquaculture, cause rapid mortality in fish hatcheries. Despite their significance, knowledge about these parasites in farmed fish in South America, especially in native species that are currently being domesticated for aquaculture, remains limited. This study morphologically characterized the Trichodinid species that are infecting Pacific fat sleeper (*Dormitator latifrons*) larvae and evaluated their impact on larval rearing. Four pairs of broodstock were induced with GnRHA implants and placed in tanks containing 200 L of freshwater, with a water temperature of 28 ± 1.0 °C and a dissolved oxygen level of 4.00 ± 1.23 mg L^−1^, with partial water exchanges being performed daily. The larvae hatched 7 to 8 h after fertilization and were transferred to tanks containing water with the same quality parameters. Twelve hours post-hatching, the presence of *Trichodina* was observed. Every 24 h, 60 larvae per tank (*n* = 180 per treatment) were sedated, and larval wet mounts were prepared, air-dried at room temperature, and impregnated with silver nitrate. Infection parameters and daily mortality were calculated. *Trichodina* was observed to parasitize the pelvic fins, caudal fins, and heads of fish larvae, which showed lethargy and erratic swimming movements. The *Trichodina* species showed a daily increase in the infection parameters, and a 58% rate of larval mortality was observed at the fourth day post-hatching (dph) in the infected tanks. In captivity, *D. latifrons* larvae typically survive up to 7 days post-hatching (dph) before reaching their point of no return due to the lack of adequate diet and feeding regimes. However, our study indicates that *Trichodina* infestation accelerates mortality, causing infected larvae to die more quickly than uninfected ones.

## 1. Introduction

The genus *Trichodina,* Ehrenberg, 1830, is a group of ciliated protozoan parasites commonly found in fish [1,2]. This group is characterized with a highly developed basal adhesive disc, a dorsal area of spiral-shaped cilia that gives it movement, and a dorsoventral ring [3]. The direct life cycle of trichodinids in natural environments would lead to high levels of infection, but healthy animals can overcome high parasitic burdens [4]. However, when the balance of host/parasites/environment is affected due to biotic and abiotic factors, such as a lack of nutrients, low water quality, and the presence of infectious diseases, trichodinids might multiply, leading to significant skin lesions and produce outbreaks [5]. In adult fish, the genus *Trichodina* causes episodes of lethargy, mainly by affectations in the gills [6], while in hatcheries, the infestations cause rapid mortalities [7].

South American continental fish aquaculture is dominated by exotic species, mainly Nile tilapia (*Oreochromis niloticus* and, to a lesser extent, *Cyprinus carpio)* and rainbow trout (*Oncorhynchus mykiss*) [8]. However, in recent years, native species have gained importance [8]. The Pacific fat sleeper, *Dormitator latifrons* (Richardson, 1844), is an euryhaline subtropical fish species that is native to the eastern Pacific Basin [9] and is considered an excellent candidate for aquaculture diversification in Latin America. This species adapts easily to the conditions of captivity [10], having high growth rates and apparent resistance to diseases [11], although the latter aspect remains underexplored. 

A significant limitation for the establishment of *D. latifrons* aquaculture is the absence of a closed larviculture protocol that would enable culture without relying on the extraction of juveniles from the wild [10,11]. Currently, Pacific sleeper larvae in captivity survive for only up to 7 days post-hatching (dph) [12,13], which represents the main bottleneck for their culture. Additionally, the incidence of pathogens and diseases—a critical issue for larviculture of aquaculture species—has been largely overlooked in the case of *D. latifrons*. During a study focused on the production of *D. latifrons* larvae under laboratory conditions, we detected the presence of ectoparasitic protozoa. Therefore, the objective of this study was to provide the first report of *Trichodina* infection in the larvae of *D. latifrons*, along with the identification, morphological characterization, and assessment of the infection parameters of two *Trichodina* species.

## 2. Materials and Methods

### 2.1. Sampling

Four pairs of broodstock of *D. latifrons* were selected from a fattening farm that depends on the supply of wild juveniles from the natural environment. Females had an average body weight of 760 ± 20.6 g and an average total length of 32 ± 2.0 cm, and the four males had an average weight of 808.3 ± 27.3 g and an average total length of 27.3 ± 2.2 cm. Each pair of broodfish were placed for 6 days in an individual tank with 300 L of freshwater (2.10 ± 0.03 g L^−1^). The husbandry conditions were as follows: a water temperature of 28 ± 1.0 °C, a photoperiod of 12 h light:12 h dark—simulating the natural photoperiod of the tropical zone, where *D. latifrons* is native—, a pH of 7 ± 0.5, an ammonia concentration of 0.79 mg L^−1^, a dissolved oxygen level of 4.00 ± 1.23 mg L^−1^ (saturation: 90%), aeration that was supplied with diffuser stones, and a partial exchange (30%) of water being performed daily. A dissolved oxygen level of 4 mg L^−1^ is enough for *D. latifrons* because this is an air-breathing species, capable of surviving in waters with low oxygen concentrations [14]. The water used for the ponds and tanks came directly from the river. The broodstocks were not given any parasitic treatment and were fed at 2% of their biomass per day with balanced shrimp feed (22% protein, Feedpac^®^ growth, Agripac, Guayaquil, Ecuador). On the seventh day, the fish were induced to spawn with one implant of 75 µg of gonadotropin releasing-hormone agonist (GnRHA) from Atlantic salmon [12], and each pair of broodfish was placed in 200 L tanks that were filled with water with the same quality parameters. 

After 24 h, both the female and male individuals matured, and the gametes were released. The broodstock were removed after natural fertilization and the eggs were left in the tank until hatching. The larvae hatched at 7 to 8 h after fertilization. From each tank (1 tank per pair of broodstock), an average of 850,000 larvae were transferred to three further tanks with 200 L of freshwater (2.1 ± 0.03 g L^−1^) with the same quality parameters as the water housing the broodstock fish. From 3 days post-hatching (dph) onwards, the larvae were fed *Brachionus* sp. rotifers twice daily to maintain a density of 10 rotifers mL^−1^, along with *Tetraselmis* sp. Within the first few hours after hatching, the protozoa of the genus *Trichodina* were observed in all three of the tanks containing larvae from the same pair of breeders. Sixty larvae per day from each tank were checked for a subsequent parasitological analysis.

### 2.2. Parasitic Characterization

The larvae were sedated every 24 h with 2−phenoxy−ethanol diluted in freshwater (0.7 mg L^−1^) and larval wet mounts were prepared and examined under a microscope, the Olympus BX53 (Olympus America Inc., Center Valley, PA, USA). When parasites were present, the wet mount was air-dried at room temperature and impregnated with Klein’s 2% silver nitrate for 8 min and washed in distilled water, followed by exposure to ultraviolet light for 20–25 min to study the details of the adhesive disc [15,16]. Morphometric measurements of the parasites were taken in accordance with Lom [17] and Van As and Basson [18]. The measurements are given in micrometer (μm). In each case, the minimum and maximum values are also provided, followed in parentheses by the arithmetic mean and standard deviation. All the slides were analyzed with a microscope, and images (300 dpi) were captured with a digital camera AmScope^®^ (United Scope LLC., Irvine, CA, USA). Schematic drawings of the denticles, as proposed by Van As and Basson [18], were made with the aid of a camera lucida attached to a microscope. 

### 2.3. Infection Parameters

The infection parameters (prevalence, mean abundance, and mean intensity) were calculated every 24 h, according to Bush et al. [19]. Briefly, prevalence (*P*) is the proportion of hosts infected with a particular parasite species, calculated by dividing the number of infected hosts (*nih*) by the total number of hosts screened (*N*).
$$P=\frac{nih}{N}\times 100$$

Mean abundance (*MA*) is the total number of individuals of a particular parasite species (*Tnp*) divided by the total number of hosts screened (*N*), regardless of whether they are infected.
$$MA=\frac{Tnp}{N}$$

Mean intensity (*MI*) refers to the total number of individuals of a particular parasite species (*Tnp*) divided by the number of hosts that are infected with that species (*n*).
$$MI=\frac{Tnp}{n}$$

### 2.4. Survival of Larvae

The daily survival rate (*S*) was calculated using the following equation: *S* = 100 × (*s*1/*s*), where *s*1 is the daily larval population and *s* is the initial larval population. The estimate of *s*1 and *s* were made using the traditional volumetric method. A statistical analysis for daily survival rate was done with Statistica v.6 software (Stat Soft. Inc., Tulsa, OK, USA). To determine significant differences in daily survival among the infected and the uninfected larvae tanks, an analysis of variance (ANOVA) factorial was performed, after checking the assumptions of normality of the residues and the homogeneity of variances. In the case of finding a difference, the Fisher post-hoc test was used. *p* values < 0.05 were considered statistically significant.

## 3. Results

### 3.1. Parasitic Characterization

Infected larvae showed lethargy and erratic swimming from the first day after hatching (dph), and ectoparasites of the genus *Trichodina* were observed in the pelvic fins, caudal fins, and heads of the *D. latifrons* larvae (Figure 1). Based on their morphology and morphometry, two species of ciliates were identified: *Trichodina acuta* and *T. compacta*.

*Trichodina acuta* Lom (1958)

The description of *T. acuta* was based on a sample size of 48 specimens. This species exhibited a mean body diameter of 72.59 ± 4.54 µm, with a range of 62.81–79.86 µm. The concave adhesive disc measured 54.36 ± 3.00 µm in diameter (range: 47.32–58.29 µm), surrounded by a marginal membrane that was finely striated and measured 3.23 ± 0.36 µm in width (range: 2.70–3.99 µm). All the detailed morphometric measurements are provided in Table 1. The central area of the adhesive disc was not well defined, and the center of the central circle lacked darker spots (Figure 2a). The nuclear apparatus featured a horseshoe-shaped and oval macronucleus (Figure 2b). The blade of the denticle was narrow and well developed, and the apex did not extend beyond the y + 1 axis. The distal surface sloped anteriorly toward the apex and the distal point was tangent to the y axis, ending in a tip that never touched the anterior denticle (Figure 2c). The posterior margin of the blade exhibited a sickle shape, while the anterior margin was smooth with a slight slope extending slightly beyond the y + 1 axis. The apophysis was well-developed and angular. The central part of the denticle, from the section connecting with the blade to the section connecting with the ray (above and below the x axis), measured 2.71 ± 0.22 µm (2.30–4.46 µm). This central part extended halfway between the y axis and the y − 1 axis. The rays were robust and mostly straight, running parallel to y axis or slightly directed posteriorly, terminating in a sharply pointed end (Figure 2c). The ray apophysis was prominent. The number of radial pins per denticle was recorded as 7.90 ± 0.56 µm (range: 7.00–9.00). A comparison of all the morphometric characteristics of *T. acuta* recorded in this study, the original description, and the specimens collected in South America are provided in Table 1. 

*Trichodina compacta* (Van As & Basson 1989).

The description of *T. compacta* was based on a sample size of 52 specimens. This species exhibited a mean body diameter of 52.18 ± 5.37 µm, with a range of 47.40–58.00. The concave adhesive disc measured 45.53 ± 4.47 µm in diameter (range: 39.32–49.92 µm), surrounded by a marginal membrane that was finely striated and measured 3.90 ± 1.10 µm in width (range: 2.03–4.93 µm). All the detailed morphometric measurements of this species are provided in Table 2. The central area of the adhesive disc was well-defined, containing an irregular central circle with darker spots, frequently supporting the edges of the rays of the denticles (Figure 2d). The area between the denticles’ ray tips and the central area formed a dark-stained ring. The nuclear apparatus displayed a horseshoe-shaped and oval macronucleus (Figure 2e). The blade was broad and squat, occupying the portion of the sector between the y and the y + 1 axes. The distal surface was flat and ran parallel to the border membrane, with the distal point remaining tangent to the y axes. The tangent point was blunt and did not make contact with the preceding denticle (Figure 2f). The posterior margin of the blade was sickle-shaped up to the height of the curve. The anterior margin exhibited a short curvature on the distal surface, followed by a slight slope that exceeded the y + 1 axis, followed by a slit and then the well-developed and angular apophysis. The central part of the blade extended before halfway point between the y axis and the y − 1 axis, with the tip of the central part reaching halfway past the y axis. The ray was short, stout, and directed anteriorly, but did not extend beyond the y + 1 axis. The ray was slightly curved, with a rounded point, and the apophysis was prominent. A comparison of all the morphometric characteristics of *T. compacta* recorded in this study, the original description, and specimens collected in South America are provided in Table 2. 

### 3.2. Infection Parameters

The two *Trichodina* species showed similar infection parameters (prevalence, mean abundance, and mean intensity), reaching prevalences above 58% at 4 dph (Table 3). Larval survival in the infected tanks was significantly lower than in the uninfected tanks (F _(3,440)_ = 97.63; *p* < 0.001). Survival in the infected tanks showed significant differences to the uninfected tanks from 2 dph (F_(9,440)_ = 1427.3; *p* < 0.001) (Figure 3).

## 4. Discussion

*Trichodina acuta* and *T. compacta* were identified infesting larvae of *D. latifrons*. The taxonomic characteristics of *T. acuta* in this study agreed with the descriptions of Lom [17,23]. The organisms from this study presented the following characteristics that agree with the species’ identification: (1) the periphery of the central circle is not well defined; (2) the center of the central circle lacks darker spots; (3) the rays never reach the periphery of the central circle, and there is a distinct gap between these; and (4) the blades well developed and distinctly sickle-shaped, and the rays have sharp points. However, the morphometrics of the organisms in this study, despite being within the ranges (Table 1), showed average values higher than those reported by Basson, Van As, and Paperna [24], Özer and Erdem [25], Dove and O’Donoghue [26], Kibria et al. [27], Maceda-Veiga et al. [28], Wang et al. [29], Yuryshynets et al. [30], de Oliveira Furtado et al. [20], and Vara et al. [31] but similar to that reported by Basson and Van As [32], Drobiniak et al. [33], Zhan et al. [34], and Wang et al. [35]. Several authors have mentioned the high morphological variation of *T. acuta* and postulated that this plasticity in the morphology is associated to variations in the environmental factors or are related to the season of the year [18,24,35,36,37].

In this study, *T. acuta* is reported for the first time infesting an endemic tropical fish species. *Trichodina acuta* was originally described from five species of freshwater fishes (*C. carpio*, *Perca fluviatilis*, *Sander lucioperca* (Syn. *Lucioperca lucioperca*), *Leucaspius delineates*, and *Rhodeus sericeus*) and in the skin of tadpoles in Czech Republic [17]. Since then, *T. acuta* has been found worldwide due to translocations of its hosts [24]. In South America, there have been two reports of *T. acuta*, both from Brazilian waters. The first report was on freshwater ornamental fishes, *Xiphophorus maculatus*, *X. helleri*, *Poecilia sphenops*, *Beta splendens*, and *Carassius auratus* [38], whereas the second was from calanoid copepods [20]. Apart from Brazil, there are scant reports of *Trichodina* species in South America. This does not necessarily indicate an absence of this parasite, but rather a lack of management and control in aquaculture systems. Given the health issues in fish associated with the presence of *Trichodina* species, our work underscores the need for the monitoring studies of this species and assessing its effects on hosts, which lead to economic losses.

The morphological characteristics of *T. compacta* identified in this study, both of their shape and measurements, are similar to those described by Van As and Basson [18]. *Trichodina compacta* was originally described as *T. acuta* by Lom [17]. However, Van As and Basson [32] separated *T. acuta* into two species: *T. acuta* and *T. compacta*. These species differ in size, the characteristics of the adhesive disc, and the shape of the denticles. The organisms from this study presented the following characteristics that agree with *T. compacta*: (1) a well-defined and clear central circle with dark spots; (2) the ray is closely associated with central circle; (3) the tips of rays are rounded; and (4) the denticles have a broad and robust blade. Our specimens showed a mean number of denticles higher than in other studies (Table 2). This increase in the number of denticles can increase the adhesion of the parasite to the small larval stages of fish. Ogut and Altuntas [39] mention that morphological changes occur during binary fission in response to a shortage of food, where they increase the number of denticles to be able to adhere to the host.

*Trichodina compacta* has been reported in several studies infesting larvae and adults of *Oreochromis niloticus* in Brazil [21,22,40,41], but this is the first record of *T. compacta* in another country of South America, and it is the first report of *T. compacta* infesting an endemic species in this region. Apparently, these species of *Trichodina* entered Ecuador with the introduction of fish species of aquacultural interest. Basson & Van As [32] mentioned that *T. compacta* and *T. acuta* colonized other hosts and localities worldwide (Africa, Europa, and Asia) due to the introduction and commercialization of tilapia and common carp around the world. Common carp and tilapia were introduced to South America in 1859 [42] and the mid-20th century [43], respectively. Furthermore, Valladão et al. [8] mentioned that this is one of the main groups of pathogens in farmed fish around the world, also affecting South American hosts. 

Both of the *Trichodina* species showed an increase in prevalence, mean abundance, and mean intensity from the first dph larval development. This result could be related to the imbalance in the parasite/host/environment relationship. The poor development of the larvae’s skins, which act as protective barriers for the fish, may be the cause of this increase. In this sense, Valladão et al. [44] found a similar result in the larvae of *Prochilodus lineatus* that were infested with *T. heterodentata*. Maciel et al. [45] mentioned that even when the levels of *Trichodina* infection in fish larvae were low, the absence of scales on the skin may cause a greater effect than in fully developed fish. Trichodinids, when firmly fixed on the host, create circular suction movements on the surface of the epithelium, causing serious damage to the tissue, which favors the entry of secondary infections [4]. Since parasitosis was observed exclusively in the larvae of a pair of broodstock, it is most likely that the infection originated from this broodstock. This suggests that the broodstock may have been the primary source of the parasites, potentially transmitting the infection to the larvae during spawning or the early developmental stages. These findings underscore the importance of screening and managing broodstock health to prevent the vertical transmission of parasites and to protect larval populations. 

Due to the absence of clinical signs of infection in the broodfish, no chemical prophylactic treatment was performed. Therefore, based on our preliminary results, we suggest the implementation of a standardized protocol to remove parasites before the induction process. The use of water from rivers without prior treatment in most *D. latifrons* culture systems can increase the probability of these parasites appearing, especially in organisms with weakened immune systems that are not fully adapted to captive conditions. Basson and Van As [4] mentioned that *Trichodina* transmission is direct, from fish to fish, so the appearance of *Trichodina* in the larvae of *D. latifrons* could be easily reduced with preventive sanitary measures and the application of quarantine protocols.

Regarding the *D. latifrons* larvae infested with *Trichodina*, we observed erratic swimming movements and lethargy, indicating stress and impaired health. Typically, *D. latifrons* larvae in captivity only survive up to 7 dph [12,13], reaching their point of no return due to the lack of an appropriate diet and feeding regime for this critical developmental stage [13]. However, our findings suggest that *Trichodina* infestation accelerates mortality in *D. latifrons* larvae, leading to faster death compared to the mortality pattern observed in uninfected larvae. This highlights the severe impact of parasitic infestations on the already limited survival of this species in captivity, emphasizing the need for better parasite management and nutritional support during early larval development. 

## 5. Conclusions

This study presents the first report of parasites infesting *D. latifrons* larvae. The infection parameters of *T*. *acuta* and *T. compacta* indicate their potential to expand their distribution range, likely facilitated by the movement of freshwater farmed fish, like tilapia and carp, that are associated with aquaculture practices. Larval mortality in the tanks that were infested with *Trichodina* was significantly higher than in the non-infested tanks, suggesting that these protozoans could negatively affect the larval rearing and the survival of *D. latifrons*. The rapid increase of *T. acuta* and *T. compacta* in the larval cultures, combined with the observed lethargy, erratic movements, and mortality of the parasitized organisms, highlights the threat that these parasites pose. The introduction of pathogens like the trichodinids reported in this study into South America and their potential impact on endemic species should be thoroughly assessed to implement the necessary sanitary measures in aquaculture systems, mainly in unstudied native fishes. Future efforts to improve *D. latifrons* larval culture should prioritize broodstock disinfection.

## Figures and Tables

**Figure 1 animals-14-03037-f001:**
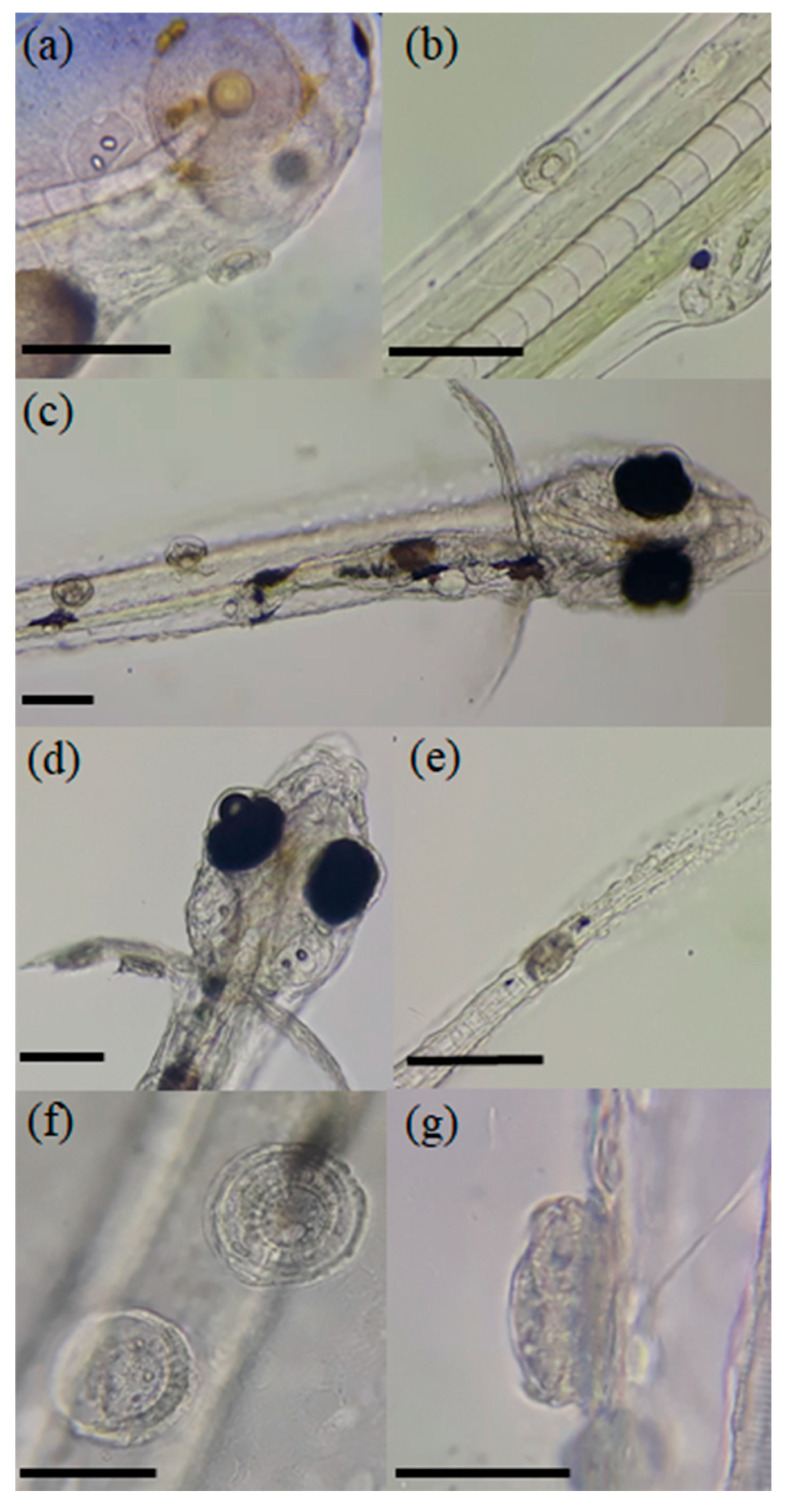
*Trichodina* infesting larvae of *Dormitator latifrons*. (**a**) Head surface of 2-day-post-hatch, infested larvae (dph); (**b**) dorsal body surface of 2 dph, infested larvae; (**c**) dorsal view of 6 dph larvae infested with *Trichodina*; (**d**) trichodinids adhered to the fins; (**e**) tail; (**f**) frontal view of *Trichodina*; (**g**) lateral view of *Trichodina*. Scale 100 µm (**a**–**e**), scale 50 µm (**f**,**g**).

**Figure 2 animals-14-03037-f002:**
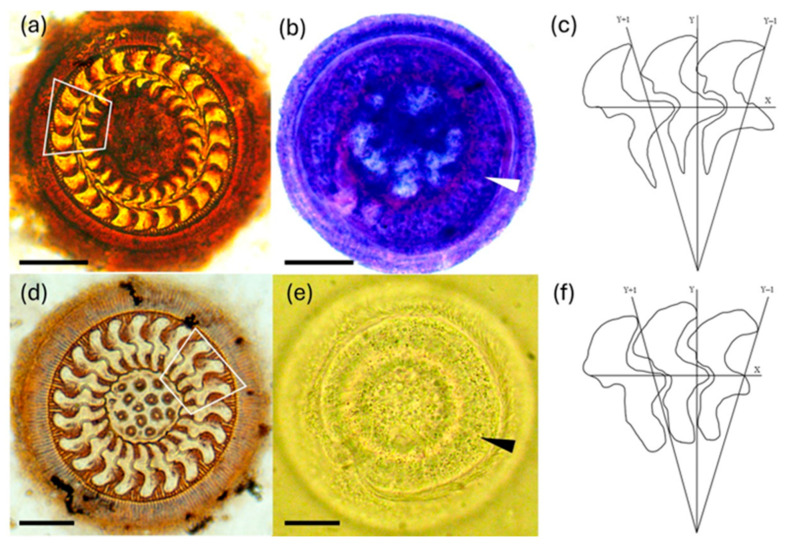
Trichodinids infesting *Dormitator latifrons* larvae. (**a**) Silver nitrate-impregnated adhesive disc of *Trichodina acuta*; (**b**) the macronucleus (arrowhead) of *T. acuta* stained by Giemsa; (**c**) a schematic drawing of the denticles of *T. acuta* (white square); (**d**) silver nitrate-impregnated adhesive disc of *Trichodina compacta*; (**e**) a macronucleus (arrowhead) in the flesh of *T. compacta*; (**f**) a schematic drawing of the denticles of *T. compacta* (white square). Scale bars: a, b = 20 µm; d, e = 12 µm.

**Figure 3 animals-14-03037-f003:**
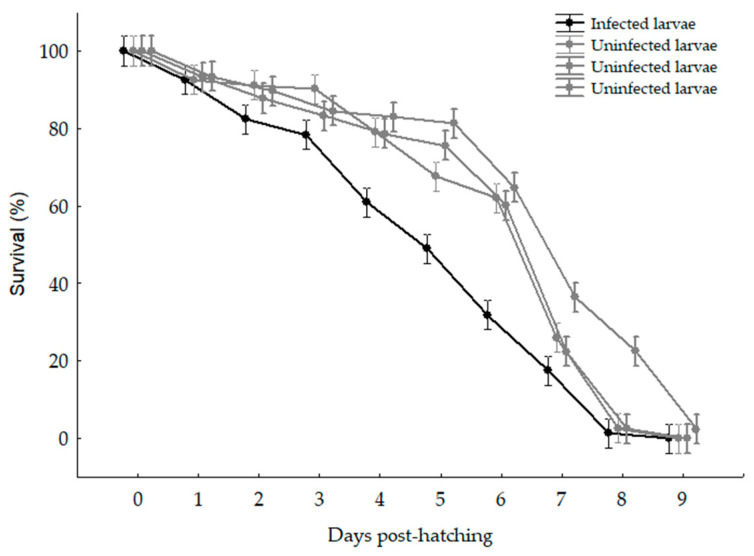
Daily mortality of larvae of *Dormitator latifrons* in tanks with larvae that were infested with *Trichodina* (line black) and tanks with larvae that had no parasites (line gray).

**Table 1 animals-14-03037-t001:** Morphometric (measurements in µm) comparison of *Trichodina acuta*, Lom, 1961, obtained in the present study, along with the original descriptions and other registers of hosts that were collected in South America. The data are presented as the arithmetic mean ± standard deviation (minimum-maximum values).

Characters	Lom (1961) [17]	Lom (1961) [17]	de Oliveira Furtado et al., 2020 [20]	Present Study
	Bohemia, Czech Republic	Benesov, Czech Republic	Brazil	Ecuador
Host	*Cyprinus carpio*, *Perca fluviatilis*, *Luciopera luciopera*, L. *delineatus*, *Rhodeus sericeus*	*Gobio holurus*	*Calanoid copepods*	*Dormitator latifrons*
Body diam.	50–86	84–110	42.7 ± 2.9 (34.5–46.3)	72.6 ± 4.5 (62.8–79.9)
A.d. diam.	30–66	63–85	42.7 ± 2.6 (29.1–38.2)	54.4 ± 3.0 (47.3–58.3)
B. m. width	3.5–5	3.5	3.6 ± 0.4 (3.0–4.5)	3.2 ± 0.4 (2.7–4.0)
D.r. diam.	18–40	18–40	18.8 ± 1.4 (15.0–20.7)	28.2 ± 2.8 (24.4–33.4)
C.c. diam.	9–12	9–12	7.8 ± 1.2 (3.8–7.8)	11.1 ± 1.3 (9.3–13.2)
Denticle number	15–23	25–30	19 (16–20)	23.4 ± 1.1 (22–25)
R.p./d	9–11	9–11	12 (7–14)	7.9 ± 0.6 (7–9)
Denticle span	10–11	10–11	10.1 ± 0.8 (8.2–11.7)	10.2 ± 0.4 (9.4–11.9)
Denticle length	–	–	5.8 ± 0.9 (3.1–7.3)	7.7 ± 0.5 (6.5–8.5)
Blade length	4–7	4–7	4.0 ± 0.4 (3.1–4.8)	4.5 ± 0.3 (4–5.2)
C.p. width	3–4	3–4	2.4 ± 0.3 (1.5–3.0)	2.7 ± 0.2 (2.3–4.5)
Ray length	4.5–6	5	4.1 ± 0.5 (3.0–5.0)	3.0 ± 0.2 (2.6–3.9)

Abbreviations: Body diam.: body diameter; A.d. diam.: adhesive disc diameter; B.m. width: border membrane width; D.r. diam.: denticle ring diameter; C.c. diam.: central circle diameter; R.p./d: radial pins per denticle; C.p. width: central part width.

**Table 2 animals-14-03037-t002:** Morphometric (measurements in µm) comparison of the *Trichodina compacta*, Van and As (1989), obtained in the present study, along with the original descriptions and other registers of hosts that were collected in South America. The data are presented as the arithmetic mean ± standard deviation (minimum-maximum values).

Characters	Van & As (1989) [18]	Ghiraldelli et al. (2006) [21]	Valladão et al. (2016) [22]	Present Study
	South Africa	Brazil	Brazil	Ecuador
Hosts	*Several hosts*	*Oreochromis niloticus*	*Oreochromis niloticus*	*Dormitator latifrons*
Body diam.	45.0 ± 3.6 (37.9–55.5)	50.8 (31.0–71.0)	49.6 ± 6.5 (36.3–60.7)	52.2 ± 5.4 (47.4–58.0)
A. d. diam.	38.1 ± 3.5 (30.9–48.4)	32.8 (19.0–40.0)	49.6 ± 6.5 (36.3–60.7)	45.5 ± 4.5 (39.3–49.9)
B. m. width	2.7 ± 4.5 (3.5–0.4)	5.6 (2.0–8.0)	4.5 ± 0.5 (3.4–5.2)	3.9 ± 1.1 (2.0–4.9)
D.r. diam.	23.2 ± 2.6 (18–30.1)	21.9 (16.0–32.0)	25.2 ± 4.5 (16.0–32.4)	25.1 ± 2.6 (22–28.8)
C.c. diam.	11.7 ± 2.1 (6.5–17.1)	8.5 (6.0–10.0)	13.0 ± 3.3 (6.3–19.1)	15.9 ± 1.7 (13.2–17.5)
Denticle number	20 (18–22)	17.0 (15.0–19.0)	19.4 ± 1.5 (17.0–22.0)	22.2 ± 0.7 (21.0–23.0)
R.p./d	10 (8–11)	7.0 (6.0–8.0)	9.2 ± 1.3 (6.0–11.0)	7.9 ± 0.5 (7.0–9.0)
Denticle span	–	13.7 (10.0–19.0)	11.1 ± 0.9 (91.0–12.8)	11.6 ± 0.5 (10.2–13.4)
Denticle length	6.9 ± 0.7 (5.5–11.3)	9.8 (6.0–19.0)	8.0 ± 1.1 (4.40–10.3)	5.1 ± 0.4 (4.2–5.9)
Blade length	3.8 ± 0.4 (2.8–4.5)	3.5 (1.0–6.0)	4.0 ± 0.5 (3.1–4.8)	4.7 ± 0.5 (4.1–5.9)
C.p. width	2.7 ± 0.5 (1.9–3.8)	1.9 (1.0–3.0)	3.0 ± 0.4 (2.3–4.0)	3 ± 0.4 (2.1–4.1)
Ray length	3.2 ± 0.4 (2.2–4.1)	5.9 (3.0–7.0)	4.1 ± 0.4 (3.4–5.0)	2.7 ± 0.5 (1.2–3.5)

Abbreviations: Body diam.: body diameter; A.d. diam.: adhesive disc diameter; B.m. width: border membrane width; D.r. diam.: denticle ring diameter; C.c. diam.: central circle diameter; R.p./d: radial pins per denticle; C.p. width: central part width.

**Table 3 animals-14-03037-t003:** Prevalence (*P*), mean abundance (*MA*), and mean intensity (*MI*) of *T. compacta* and *T. acuta* infesting larvae of *Dormitator latifrons*. Mean ± D.E. (standard deviation). *MA* and *MI* are expressed as the number of individuals.

Day Post-Hatch	*Trichodina compacta*			*Trichodina acuta*		
	*P* (%)	*MA*	*MI*	*P* (%)	*MA*	*MI*
1	3 ± 1.67	0.03 ± 0.18	1.00 ± 0.00	2 ± 0.88	0.03 ± 0.09	1.00 ± 0.00
2	7 ± 1.67	0.07 ± 0.25	1.00 ± 0.00	6 ± 1.71	0.05 ± 0.12	1.33 ± 0.00
3	22 ± 3.33	0.22 ± 0.41	1.10 ± 0.00	20 ± 1.66	0.22 ± 0.20	1.00 ± 0.00
4	63 ± 4.17	0.63 ± 0.86	1.50 ± 0.45	58 ± 0.88	0.60 ± 0.42	1.58 ± 0.52
5	55 ± 4.17	0.55 ± 0.87	1.48 ± 0.35	51 ± 1.71	0.53 ± 0.42	1.53 ± 0.42
6	2 ± 12.33	0.02 ± 0.13	1.00 ± 0.00	1 ± 0.38	0.02 ± 0.06	1.00 ± 0.06

## Data Availability

Data are contained within the article.

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
