# Peer review of "The Effect of Parasitization by Trichodinid Ciliates on the Mortality of Cultured Pacific Fat Sleeper larvae (Dormitator latifrons)"

_animals, 2024, doi:10.3390/ani14203037_

Round 1
Reviewer 1 Report
Comments and Suggestions for Authors
The authors discovered that Pacific fat sleepers are infected with Trichodinids first and identified two species. This is one achievement, but Trichodinids are already known that are widespread parasites and can cause the mortality of larvae and fry. The mortality reached 100% in both the infected and uninfected tanks, and the difference in survival time was only a few days, so it seems that there was a problem with the experimental environment. The authors present water quality as the cause, but you did not compare water quality in this experiment. The authors discuss methods to deal with the problem, but all of the evidence is based on citations and has not been confirmed by experiment. In addition, there are some parts where the Materials and Methods, and Results are insufficiently explained. However, these results may be useful for the development of aquaculture techniques for Pacific fat sleepers in the future. For the above reasons, I would like to recommend that the authors rewrite the manuscript and submit to "communications" or someone. Details are described below.
Lines 59-64: Please clarify the relationship between the background of this research (larval rearing for this species is the bottleneck that ---) and the objective (Therefore, objective of this work was to identify and characterize morphologically ---). I think that merely examining the morphology of known Trichodina is insufficient as the objective of an original article.
Lines 72-73: Please explain why the photoperiod was set to 12L12D.
Lines 79-80: Please provide a brief explanation: hormone concentrations, number of days until final maturity of both sexes, artificial or natural fertilization, etc.
Lines 83-84: Please describe the methods taken to prevent the loss of eggs.
Line 88: Does this mean that only one of the four tanks was infected?
Lines 88-89: Please briefly explain the method of "parasitological analysis".
Line 91: Is this equation correct? When you apply the values, won't m become negative?
Line 108: Please briefly explain the contents of "recommendations".
Line 116: Please briefly explain whow to make "prevalence, mean abundance, and mean intensity".
Figure 2: Please indicate which part of the figure is the "nuclear apparatus". Is it okey to use the same scales to Figures-a and -b, and -d and -e? Please indicate which part of Figure-a is Figure-c. Please indicate that the nucleus in Figure-e, because it is not clear. Please explain why the staining method was changed between Figures-b and -e.
Line 194: Please describe the statistical method.
Lines 265-267: Please add a discussion why one tank was infected but the other three tanks were not infected in this experiment.
Lines 282-283: In this experiment, the mortality rate in the non-infected tanks also reached 100% in a few days. It is true that the mortality in the infected tank increased quickly, but I don't think you can say that only this tank had a high mortality.
Lines 283-290: Since you did not conduct any experiments on treating infected fish in this study, I don't think you can discuss this point.
Lines 300-302: Since you did not confirm this in the experiment, I think it would be better to remove it.
Author Response
Comment: I would like to recommend that the authors rewrite the manuscript and submit to "communications" or someone.
Response: We agree
Comment Lines 59-64: Please clarify the relationship between the background of this research (larval rearing for this species is the bottleneck that ---) and the objective (Therefore, objective of this work was to identify and characterize morphologically ---). I think that merely examining the morphology of known Trichodina is insufficient as the objective of an original article.
Response: The paragraph was modified to clarify the relationship. The objective was changed to “The objective of this study was to provide the first report of Trichodina infection in larvae of D. latifrons, along with the identification, morphological characterization, and assessment of the infection parameters of two Trichodina species”
Comment Lines 72-73: Please explain why the photoperiod was set to 12L12D.
Response: A sentence –simulating the natural photoperiod of the tropical zone, where D. latifrons is native– was included.
Comment Lines 79-80: Please provide a brief explanation: hormone concentrations, number of days until final maturity of both sexes, artificial or natural fertilization, etc.
Response: The request information was included
Comment Lines 83-84: Please describe the methods taken to prevent the loss of eggs.
Response: As explained in the same sentence, the eggs remained in the tank until hatching; while the broodstock were removed and placed in other tanks. Due to their small size and their characteristic of sticking to the walls of the tank, eggs are not collected, the breeders are removed from the tank once they spawn.
Comment Line 88: Does this mean that only one of the four tanks was infected?
Response: We noticed that the initial wording was confusing and did not allow to understand the management of the larvae. The wording was improved to clarify
Comment Lines 88-89: Please briefly explain the method of "parasitological analysis".
Response: The parasitological methods are detailed in section 2.2. between lines 102-113.
Comment Line 91: Is this equation correct? When you apply the values, won't m become negative?
Response: The equation is correct, but the acronym description is reversed. Corrected
Comment Line 108: Please briefly explain the contents of "recommendations".
Response: The term recommendations was removed to avoid confusion. Lom´s recommendations refers to the measurements of the body parts of the Trichodinas presented in tables 1 and 2
Comment Line 116: Please briefly explain whow to make "prevalence, mean abundance, and mean intensity".
Response: A brief explanation of how to calculate 'prevalence, mean abundance, and mean intensity' was included.
Comment Figure 2: Please indicate which part of the figure is the "nuclear apparatus". Is it okey to use the same scales to Figures-a and -b, and -d and -e? Please indicate which part of Figure-a is Figure-c. Please indicate that the nucleus in Figure-e, because it is not clear. Please explain why the staining method was changed between Figures-b and -e.
Response: We included what was requested and reviewed the scales.
Comment Line 194: Please describe the statistical method.
Response: A briefly description of statistic method was included in a new section 2.4
Comment Lines 265-267: Please add a discussion why one tank was infected but the other three tanks were not infected in this experiment.
Response: Done
Comment Lines 282-283: In this experiment, the mortality rate in the non-infected tanks also reached 100% in a few days. It is true that the mortality in the infected tank increased quickly, but I don't think you can say that only this tank had a high mortality.
Response: The sentence was rewritten to clarify
Comment Lines 283-290: Since you did not conduct any experiments on treating infected fish in this study, I don't think you can discuss this point.
Response: The discussion of this point was eliminated
Comment Lines 300-302: Since you did not confirm this in the experiment, I think it would be better to remove it.
Response: Paragraph was removed
Reviewer 2 Report
Comments and Suggestions for Authors
The manuscript descripted two trichodinid ciliates associated to larval rearing mortality of the Pacific fat sleeper. Trichodina acuta and T. compacta have been reported in the previous research, and the mortality of Pacific fat sleeper also can not be ascribed to trichodinid infection. Therefore, the scientific soundeness of this manscript is limited.
Author Response
Comments and Suggestions for Authors
The manuscript descripted two trichodinid ciliates associated to larval rearing mortality of the Pacific fat sleeper. Trichodina acuta and T. compacta have been reported in the previous research, and the mortality of Pacific fat sleeper also can not be ascribed to trichodinid infection. Therefore, the scientific soundeness of this manscript is limited.
Response: Based on the feedback from the other two reviewers, we have made significant improvements to the manuscript. In addition to describing the two Trichodina species, we now report the incidence and impact of these pathogens on the larval culture of Dormitator latifrons. This added information is crucial for understanding the challenges posed by these parasites and provides valuable insights for the development of D. latifrons aquaculture. By addressing both the identification and effects of these pathogens, the manuscript offers a more comprehensive perspective that will be beneficial for future research and management practices in larval culture.
Reviewer 3 Report
Comments and Suggestions for Authors
This manuscript found that the trichodinid ciliates in the specific fish species for the first time. Authors argue that this parasite could case the serve the disease during the hatching of the Pacific fat sleeper. The paper is easy to read and well understood. However, I suggest some scientific data and discussion issues should be improved before publication.
Line 29, Is the O2 concentration (4mg/l) enough for this tropic fish? Explain why?.
Line 198, please show the bar of SD or SEM in the graph of Mortality of larvae so that readers can know how significance between them.
Line 270-271, the curable drugs or reagents, to some extent, there are some published evidence (eg., Peracetic acid), and also by experienced farmers in China I have heard from, usually the CuSO4 would be helpful for alleviating this disease when the symptoms are very exhausted. Please include some literatures and discuss more to appeal more attention for researchers.
Author Response
Suggestions for Authors
This manuscript found that the trichodinid ciliates in the specific fish species for the first time. Authors argue that this parasite could case the serve the disease during the hatching of the Pacific fat sleeper. The paper is easy to read and well understood. However, I suggest some scientific data and discussion issues should be improved before publication.
Comment Line 29: Is the O2 concentration (4mg/l) enough for this tropic fish? Explain why?.
Response: Yes, it is enough because D. latifrons is an air-breathing fish and inhabits environments with low Oâ‚‚ levels. This argument was included in methodology
Comment Line 198: please show the bar of SD or SEM in the graph of Mortality of larvae so that readers can know how significance between them.
A: Done
Comment Line 270-271: the curable drugs or reagents, to some extent, there are some published evidence (eg., Peracetic acid), and also by experienced farmers in China I have heard from, usually the CuSO4 would be helpful for alleviating this disease when the symptoms are very exhausted. Please include some literatures and discuss more to appeal more attention for researchers.
Response: It is certainly an interesting topic that may attract the attention of other researchers. However, based on the recommendation of another reviewer, this section was removed since no treatments were tested in this study. Additionally, it is important to mention that larvae at 7 dph are barely 1 mm total length, making them very sensitive to chemicals
Round 2
Reviewer 1 Report
Comments and Suggestions for Authors
The authors have responded appropriately to the previous comments.
However, please consider making the following one correction.
Lines 100-103: Please add the meaning of “n1, n, m”.
Author Response
Thank you for your comments. The answers are detailed in the attached PDF document.
